# Identification of PECAM1 as a Prognostic Biomarker for Lung Adenocarcinoma

**DOI:** 10.3390/diagnostics15091094

**Published:** 2025-04-25

**Authors:** Shih-Sen Lin, Pei-Sung Hsu, Ying-Chu Lin, Jie-Yu You, Yung-Leun Shih, Hung-Chih Lai

**Affiliations:** 1Division of Chest Medicine, Department of Internal Medicine, Shin Kong Wu Ho-Su Memorial Hospital, Taipei 111045, Taiwan; m008932@ms.skh.org.tw; 2Department of Pulmonology Medicine, Shin Kong Wu Ho-Su Memorial Hospital, Taipei 111045, Taiwan; m006541@ms.skh.org.tw; 3Department of Hematology and Oncology, Shin Kong Wu Ho-Su Memorial Hospital, Taipei 111045, Taiwan; chuchu0419.lin@gmail.com; 4Department of Pathology and Laboratory Medicine, Shin Kong Wu Ho-Su Memorial Hospital, Taipei 111045, Taiwan; 5Graduate Institute of Pharmacology, College of Medicine, National Taiwan University, Taipei 111045, Taiwan

**Keywords:** lung cancer, NSCLC, DEGs, PECAM1, bioinformatics, immune infiltration

## Abstract

**Background**: Lung cancer continues to be one of the most fatal malignancies globally. Uncovering differentially expressed genes (DEGs) is crucial for advancing our understanding of tumor mechanisms and discovering new therapeutic targets. This study sought to identify key genes linked to prognosis and immune infiltration in lung cancer through the analysis of public gene expression datasets. **Methods**: We examined three microarray datasets from the Gene Expression Omnibus (GSE10072, GSE33356, and GSE18842) to detect DEGs between tumor and normal lung tissues. Functional enrichment was performed using Gene Ontology (GO) and Kyoto Encyclopedia of Genes and Genomes (KEGG) pathway analyses to interpret the biological relevance of these genes. Protein–protein interaction (PPI) networks were constructed via STRING and visualized using Cytoscape to screen for central hub genes. The prognostic implications of the hub genes were investigated using Kaplan–Meier Plotter and TIMER2.0 based on data from The Cancer Genome Atlas (TCGA). PECAM1 expression levels and its relationship with immune cell infiltration were further explored using UCSC Xena. **Results**: A total of 477 DEGs were consistently identified across all three datasets. Among the top 10 down-regulated hub genes, PECAM1 was significantly reduced in tumor tissues. Lower PECAM1 expression was positively associated with better first-progression survival (FPS) in lung cancer patients. This gene was particularly suppressed in lung adenocarcinoma (LUAD) and showed strong correlations with immune cell infiltration. Co-expression analysis revealed that genes linked to PECAM1 are involved in immune-related pathways. **Conclusions**: Our findings highlight PECAM1 as a potential prognostic biomarker in lung cancer, especially in LUAD. Its association with immune infiltration and patient survival supports its possible utility in early detection and as a candidate for immunotherapy development.

## 1. Introduction

Lung cancer continues to rank among the most common and deadly malignancies worldwide. Histologically, it is broadly divided into two major types: small cell lung cancer (SCLC) and non-small cell lung cancer (NSCLC). Of these, NSCLC accounts for approximately 85% of all lung cancer cases. The two most prevalent NSCLC subtypes are lung adenocarcinoma (LUAD) and lung squamous cell carcinoma (LUSC), which differ in their pathological features, clinical behavior, and treatment responses [1]. Although considerable progress has been made in the development of therapeutic approaches—such as molecular targeted agents and immune checkpoint inhibitors—many patients either do not harbor actionable genetic alterations or eventually develop resistance to these therapies, limiting long-term treatment efficacy [2,3,4,5]. This underscores the urgent need for novel biomarkers and therapeutic targets to improve diagnosis, prognosis, and treatment responsiveness.

In recent years, integrative bioinformatics approaches have played a critical role in unraveling the molecular landscape of lung cancer. For example, Varnamkhasti et al. identified a panel of genes associated with lymphatic metastasis in primary lung squamous cell carcinoma, demonstrating the power of transcriptomic profiling to predict metastatic progression [6]. Similarly, Alipour et al. utilized a combination of transcriptomics and functional analysis to highlight the significance of LINC00894, YEATS2-AS1, and SUGP2 as potential biomarkers for nodal status in LUAD [7]. These studies reinforce the value of public datasets and bioinformatics tools in discovering clinically relevant molecular targets.

Against this backdrop, our study aimed to identify and characterize potential hub genes involved in LUAD progression and immune modulation by leveraging publicly available transcriptomic datasets. Among the candidates identified, Platelet Endothelial Cell Adhesion Molecule-1 (PECAM1/CD31) emerged as a hub gene with notable prognostic relevance. PECAM1 is a cell adhesion molecule expressed predominantly on endothelial cells and plays critical roles in angiogenesis, immune regulation, and the maintenance of endothelial barrier integrity [8]. Previous studies have implicated PECAM1 in tumor cell migration, proliferation, and anchorage-independent growth—hallmarks of metastatic potential [9]. Moreover, PECAM1 is closely associated with angiogenesis, which is another essential process for tumor growth and metastasis [10]. The expression of PECAM1 has also been correlated with the proliferation and metastasis of melanoma and breast cancer cells [11]. It has known roles in the immune response to tumors; the expression of PECAM1 can impact the infiltration of immune cells into the tumor microenvironment, which is a crucial factor in the immune surveillance of cancer [10].

Our findings revealed that PECAM1 is significantly down-regulated in lung cancer and positively associated with first-progression survival (FPS) among patients. Functional analyses also demonstrated strong associations between PECAM1 and genes involved in immune regulation and inflammatory signaling. Additionally, PECAM1 expression was inversely correlated with tumor purity and positively associated with the infiltration of multiple immune cell subsets, including T cells, B cells, and macrophages. These observations suggest that PECAM1 may serve not only as a prognostic biomarker but also as a promising candidate for immunotherapeutic targeting in LUAD. Together, our research highlights the multifaceted role of PECAM1 in lung cancer and supports its potential clinical utility.

## 2. Materials and Methods

### 2.1. Data Sources

Three datasets—GSE10072, GSE33356, and GSE18842—sourced from the Gene Expression Omnibus (GEO; https://www.ncbi.nlm.nih.gov/geo/ (accessed on 16 July 2024)) public database were utilized to conduct a comparative RNA analysis between human lung cancer and normal tissues [12,13,14]. GSE10072: This dataset was generated using the Affymetrix Human Genome U133 Plus 2.0 Array (GPL96) platform and contains 58 lung adenocarcinoma tissue samples and 49 matched normal lung tissues, obtained from surgical resections. GSE33356: This dataset was also based on the Affymetrix Human Genome U133 Plus 2.0 Array (GPL570) platform and includes 60 primary NSCLC samples (including both LUAD and LUSC subtypes) and 60 normal lung tissue samples, derived from Japanese patients. GSE18842: This dataset was created using the Affymetrix Human Genome U133 Plus 2.0 Array (GPL570) and comprises 46 lung cancer samples (including both LUAD and LUSC) and 45 adjacent normal lung tissue samples, collected from patient biopsies. These datasets were chosen for their adequate sample size, standardized platform use, and inclusion of both tumor and matched normal samples, enabling the reliable cross-comparison and identification of consistently differentially expressed genes (DEGs) across lung cancer studies. These datasets were meticulously analyzed using various public platforms and analytical software to validate gene expression patterns and their correlations within the collected data, as presented in Figure 1.

### 2.2. Filtering Differentially Expressed Genes in Lung Cancer

The interactive online platform GEO2R (https://www.ncbi.nlm.nih.gov/geo/geo2r/ (accessed on 16 July 2024)) was used to identify DEGs in lung cancer. This platform utilizes GEOquery and limma within the R programming environment to analyze gene expression data from microarrays. Subsequent statistical analysis was performed using the limma package (Linear Models for Microarray Data), which helps to identify significant differences in gene expression [15]. We isolated DEGs between normal and lung cancer tissues by selecting those with an adjusted *p*-value < 0.01 (Benjamini–Hochberg corrected) and an absolute log fold change (|logFC|) > 1.0 for further investigation. These criteria ensured the identification of genes with both statistically significant and biologically relevant changes in expression. InteractiVenn (https://www.interactivenn.net/ (accessed on 23 July 2024)) enabled us to compare three sets of GSE datasets and identify intersecting DEGs that were either up- or down-regulated [16].

### 2.3. GO and KEGG Analysis of DEGs

ShinyGo (http://bioinformatics.sdstate.edu/go/ (accessed on 3 February 2025)) version 0.82 was utilized for gene set enrichment analysis (GSEA), which included both Gene Ontology (GO) and Kyoto Encyclopedia of Genes and Genomes (KEGG) analyses [17,18,19]. GO analysis categorizes DEGs into three functional groups: cellular components (CCs), biological processes (BPs), and molecular functions (MFs). KEGG analysis delineates signaling pathways implicated by DEGs. To define statistical significance, the following thresholds were applied consistently across all enrichment analyses: *p*-value < 0.01 and False Discovery Rate (FDR) < 0.05. These cutoffs ensured the robustness of enriched term identification. ShinyGO ranks terms based on FDR values, gene count, and fold enrichment score to elucidate the cellular functions and pathways engaged by the DEGs.

### 2.4. Investigation of Protein–Protein Interactions and Hub Genes

To identify central regulatory genes among the DEGs, we first uploaded the 334 consistently down-regulated DEGs into the STRING database (https://string-db.org/ (accessed on 24 July 2024)) to construct a protein–protein interaction (PPI) network [20,21]. The minimum required interaction score was set at 0.4 (medium confidence) to ensure biologically meaningful interactions while maintaining network connectivity. The resulting PPI network was then imported into Cytoscape v3.10.2, an open-source platform widely used for network visualization and biological interaction modeling [22]. The parameters of the multi-contrast delayed enhancement (MCODE) plug-in (version 1.5.1) (Degree Cutoff = 2, Node Score Cutoff = 0.2, K-Core = 2, and Max. Depth = 100) were utilized to search subnets. To prioritize key nodes within the network, we employed the CytoHubba plugin in Cytoscape, which provides a suite of topological analysis algorithms for hub gene identification [22,23]. Among the 12 available scoring methods, we selected the Degree centrality algorithm, which ranks nodes based on the number of direct connections (edges) a node has in the network. This method was chosen for its robustness, interpretability, and broad applicability in biomolecular network analysis. Genes with higher degree values are considered more functionally influential due to their extensive interaction profiles. The top 10 down-regulated hub genes—namely, IL6, NR4A1, FOSB, PECAM1, FOS, DUSP1, NR4A2, EGR1, ATF3, and ZFP36—were selected for further downstream analysis, including prognostic evaluation and immune correlation studies.

### 2.5. Genomic Expression and Prognosis of Hub Genes in LUAD

To investigate gene expression differences in LUAD, we utilized the UCSC Xena platform (https://xena.ucsc.edu/#overview (accessed on 17 May 2024)), which aggregates genomic and phenotypic data from 129 cohorts and over 1500 datasets spanning large-scale cancer research projects [24,25]. This resource enabled a comprehensive comparison of hub gene expression between LUAD and corresponding normal tissues, providing insight into the molecular characteristics of LUAD. We also employed the Kaplan–Meier Plotter tool (https://kmplot.com/analysis/ (accessed on 16 June 2024)) to explore the association between gene expression levels and clinical outcomes in LUAD patients. The analysis used the best auto-selected cutoff and applied Cox proportional hazards regression to ensure robust survival modeling [26,27]. This resource enabled a comprehensive comparison of hub gene expression between LUAD and corresponding normal tissues, providing insight into the molecular characteristics of LUAD [28,29].

### 2.6. TIMER2.0 Analysis of DEG Expression in Pan-Cancer and Immune Cell Correlations

To evaluate the relationship between gene expression and immune cell presence in the tumor microenvironment, we used the TIMER2.0 platform (http://timer.cistrome.org/) [30,31]. This tool enabled us to assess gene-specific correlations with various immune cell types across multiple TCGA cancer types, including LUAD. TIMER2.0 also facilitated comparisons of gene expression between tumor and normal tissues, as well as immune-related survival analyses and gene co-expression profiling.

### 2.7. Analyzing Multi-Omics Data in LinkedOmics Suite

We further explored the biological relevance of PECAM1 using the LinkedOmics database (http://www.linkedomics.org/ (accessed on 16 July 2024)), which integrates genomic, transcriptomic, and proteomic data from TCGA and CPTAC cancer cohorts [32]. Within the TCGA-LUAD dataset, we analyzed genes co-expressed with PECAM1 using Pearson correlation analysis, identifying the top 50 genes with positive and negative correlation coefficients. The results were visualized through heatmaps and volcano plots. To identify enriched biological functions associated with PECAM1 co-expression, we used the LinkInterpreter module, which performs Gene Ontology (GO) and Gene Set Enrichment Analysis (GSEA). For enrichment ranking, we applied a minimum gene set size of 3, FDR < 0.05, and 500 permutations, ensuring the robustness of statistical inference.

### 2.8. Statistical Analysis

Differential expression between LUAD and normal tissues from the TCGA database was assessed using two-tailed independent sample *t*-tests. To adjust for multiple comparisons, we applied the Benjamini–Hochberg procedure to control the false discovery rate. Survival analysis was conducted via Kaplan–Meier curves, log-rank tests, and univariate Cox regression models. The correlation between PECAM1 expression and immune infiltration levels was determined using Spearman’s rank correlation coefficient, with correlation strength interpreted as weak (|r| < 0.25), moderate (0.25 ≤ |r| < 0.75), or strong (|r| ≥ 0.75). A *p*-value less than 0.05 was regarded as statistically significant.

## 3. Results

### 3.1. Unveiling the Landscape of DEGs in Lung Cancer

To investigate DEGs in lung cancer, we obtained three microarray datasets from the GEO database: GSE10072, GSE33356, and GSE18842. We utilized volcano plots to visually represent the DEGs, where statistical significance (expressed as -log10 of the *p*-value) was plotted against the magnitude of expression change (log2 fold change) (Figure 2A–C). The datasets were categorized into normal and lung cancer tissues. In the GSE10072 dataset, which comprised 49 normal and 58 lung cancer tissues, we identified 230 up-regulated and 432 down-regulated DEGs (adjusted *p* < 0.01, |logFC| > 1.0). Similarly, in GSE33356, which included 60 normal and 60 lung cancer tissues, 455 up-regulated and 949 down-regulated DEGs (adjusted *p* < 0.01, |logFC| > 1.0) were identified. Lastly, GSE18842, with 45 normal and 46 lung cancer tissues, revealed 1416 up-regulated and 1785 down-regulated DEGs (adjusted *p* < 0.01, |logFC| > 1.0) (Figure 2D–F). A Venn diagram was used to visualize the intersection of the DEGs based on their expression trends to identify commonalities across the datasets. This analysis revealed 143 DEGs consistently up-regulated and 334 DEGs consistently down-regulated across the datasets (Figure 2E,F), enabling the pinpointing of critical genes potentially implicated in lung cancer pathogenesis and providing a foundation for further exploration of their roles and therapeutic potential.

### 3.2. Deciphering the Functional Roles of DEGs in Lung Cancer

We utilized GO analysis and GSEA to elucidate the cellular functions and pathways associated with the down-regulated DEGs in lung cancer. GO analysis (*p* < 0.01, FDR < 0.05) identified that within the CC category, these down-regulated DEGs are predominantly involved in structures such as stress fibers, contractile actin filament bundles, collagen-containing extracellular matrices, and membrane rafts and microdomains. These genes exhibited significant enrichment in extracellular matrices, external encapsulating structures, cell–cell junctions, anchoring junctions, and focal adhesions (Figure 3A). Regarding BP, the DEGs showed enrichment in crucial areas of vascular biology, such as blood vessel morphogenesis, angiogenesis, and the development of both blood vessels and the overall vasculature. This extends to tube morphogenesis, the development of the circulatory system, and the formation of anatomical structures involved in morphogenesis, alongside cell adhesion and migration processes (Figure 3B). MF analysis revealed that the DEGs were enriched in a variety of receptor activities and binding functions, including adrenomedullin receptor activity, glucocorticoid receptor binding, transforming growth factor-beta binding, and integrin binding. Additional functions included amyloid-beta binding, transmembrane receptor protein kinase activity, and roles as extracellular matrix structural constituents. Glycosaminoglycan, heparin, and cytokine binding also featured prominently (Figure 3C). Lastly, KEGG pathway analysis (*p* < 0.01, FDR < 0.05) indicated that the DEGs were enriched in pathways associated with the renin–angiotensin system; complement and coagulation cascades; and those related to diseases like malaria, rheumatoid arthritis, and diabetic complications. These DEGS also have roles in the adherent junction pathway, vascular smooth muscle contraction, hypertrophic cardiomyopathy, and the mechanisms of fluid shear stress and atherosclerosis (Figure 3D). The integration of these datasets and analytical methods provides a comprehensive view of the genetic alterations in lung cancer, contributing to our understanding of lung cancer biology.

### 3.3. Identification of Top 10 Down-Regulated Hub Genes in Lung Cancer

By using STRING on 334 down-regulated DEGs according to FDR < 0.05 and minimum strength > 0.01, a protein–protein interaction (PPI) network was established. The PPI enrichment *p*-value was <1.0 × 10^−16^, resulting in a total of 317 nodes and 1044 edges (Figure 4A). Subsequently, we used the Cytohubba plugin in the Cytoscape software to rank these DEGs, thus identifying the top 10 hub genes that play a central role in the network. These genes were interleukin 6 (IL6), nuclear receptor 4A1 (NR4A1), FosB proto-oncogene (FOSB), platelet and endothelial cell adhesion molecule 1 (PECAM1), Fos proto-oncogene (FOS), dual-specificity phosphatase 1 (DUSP1), nuclear receptor subfamily 4 group A member 2 (NR4A2), early growth response 1 (EGR1), activating transcription factor 3 (ATF3), and zinc finger protein 36 (ZFP36) (Figure 4B–D). Our comprehensive evaluation revealed that these top 10 hub genes were consistently down-regulated in lung tumor tissues across the three independent GEO datasets. This consistent down-regulation underscores the potential significance of these genes in lung cancer pathophysiology and highlights them as promising targets for further investigation in lung cancer research.

### 3.4. Assessing the Prognostic Value of Hub Genes in Lung Adenocarcinoma

In our analysis of NSCLC, we focused on the LUAD subtype, which is the most prevalent followed by LUSC [33]. Utilizing the TCGA database, we examined the expression levels of the top 10 hub genes in LUAD. Our findings indicate that the expression of these genes was significantly down-regulated in LUAD tissues (Figure 5). Further investigation using the Kaplan–Meier plotter revealed that the high expression levels of two hub genes, FOSB and PECAM1, were associated with a significantly extended first-progression survival (FPS) in lung cancer patients, with hazard ratios (HRs) of 0.63 and 0.56, respectively (Figure 6). Subtype-specific analysis further confirmed the consistent pattern of PECAM1 expression across magnoid, squamous, and bronchial tumor subtypes compared to normal tissues (Figure 7), indicating the potential of PECAM1 as a target gene for lung cancer with prognostic implications.

### 3.5. PECAM1 Was Positively Associated with NSCLC Prognosis

Our meta-analysis encompassing multiple studies revealed that PECAM1 expression was significantly reduced in both LUAD and LUSC subtypes of NSCLC. Specifically, data from seven distinct studies yielded a standardized mean difference (SMD) of −2.90 for PECAM1 in LUAD within a 95% confidence interval (CI) (Figure 8A). Similarly, data from five studies indicated an SMD of −3.93 for PECAM1 in LUSC, highlighting its consistent down-regulation in NSCLC (Figure 8B). Furthermore, a comprehensive assessment of PECAM1 across various cancer types revealed that PECAM1 was consistently expressed at low levels compared to normal tissues, including bladder, breast, esophageal, colon, cervical, renal cell, pancreatic, rectal, and endometrial cancers (Figure 9A). When examining the impact of PECAM1 expression on the overall survival rate in NSCLC subtypes, we observed a dichotomy: high PECAM1 expression was linked to longer overall survival and a lower HR (HR = 0.64) in LUAD patients, whereas in LUSC patients, elevated PECAM1 levels were associated with shorter overall survival and an increased HR (HR = 1.4) (Figure 9B,C). These findings suggest that PECAM1 may serve as a prognostic biomarker, potentially enhancing cancer prognosis in LUAD patients. This differential expression and its impact on patient outcomes highlight the necessity for subtype-specific therapeutic strategies in NSCLC.

### 3.6. Genes Positively Correlated with PECAM1 Expression Up-Regulated Immune Function in LUAD

In our analysis of the LUAD database from TCGA, we focused on genes correlated with PECAM1 and classified them based on their correlation. Fifty genes were identified with significant positive or negative correlations to PECAM1 (*p* < 0.05), as shown in Figure 10A–C. Through GO and GSEA analyses, we identified an up-regulation in MFs, such as TNF superfamily cytokine production, IL4 production, and adaptive immune response. In contrast, BPs, such as cytokinesis, RNA metabolic processes, repair, and DNA-templated transcription elongation, were down-regulated (Figure 10D). GSEA further highlighted that PECAM1-associated genes were significantly enriched in TNF superfamily cytokine production and the adaptive immune response (Figure 10E), suggesting a potential anti-cancer role of PECAM1 and an ability to improve LUAD prognosis by enhancing immune defense.

### 3.7. PECAM1 Is a Marker of Immune Infiltration in LUAD

Our investigation into the role of PECAM1 in lung cancer through TIMER2.0 analysis revealed a significant negative correlation with tumor purity (Rho = −0.427). This suggests that higher PECAM1 expression may be indicative of reduced tumor progression. Correlation analysis between immune cells and PECAM1 expression showed significant positive associations with CD8+ T cell (Rho = 0.217), CD4+ T cell (Rho = 0.161), regulatory T cell (Rho = 0.392), B cell (Rho = 0.221), and macrophage/monocytes (Rho = 0.259) infiltration (Figure 11). These findings highlight the potential of PECAM1 as a biomarker for immune cell infiltration in LUAD with prognostic value, with higher expression correlated with longer overall survival in LUAD patients. The data also suggest that the impact of PECAM1 on prognosis may stem from its role in regulating immune cell infiltration.

## 4. Discussion

Our study has identified and characterized genes with significant differential expression in lung cancer, which could serve as potential targets for future therapeutic intervention. We conducted an extensive gene analysis of lung cancer using public gene databases. Through the GEO, the extraction of data from three microarray gene expression datasets identified 477 DEGs with significant variance in lung cancer cases. Furthermore, GO analysis shed light on the BPs, MFs, and CCs associated with these DEGs and revealed significant involvement in essential biological activities. In addition, insights obtained from KEGG into the functions of these gene pathways enrich our understanding of their roles in lung cancer. Exploring protein–protein interaction networks using STRING and filtered hub genes with Cytoscape software facilitated ranking of the following top 10 hub genes for further investigation: IL6, NR4A1, FOSB, PECAM1, FOS, DUSP1, NR4A2, EGR1, ATF3, and ZFP36. These candidate genes underwent additional comparative analysis against normal tissue expression profiles within the TCGA database and thus revealed their differential expression in lung cancer. This finding enhances our understanding of the correlation between gene expression and the progression of lung cancer.

The use of bioinformatics tools in our study provided prognostic insights into key hub genes. The Kaplan–Meier plotter was instrumental in evaluating first-progression survival (FPS) by offering a deeper understanding of the prognostic relevance of these genes. We examined the expression profiles of essential candidate genes through databases such as Lung Cancer Explorer and TIMER2.0 and then contrasted their expression in cancerous versus normal tissues. LinkedOmics analysis facilitated the identification of genes with significant correlations to our hub genes, and GSEA allowed for the assessment of gene abundance within various functional sets. Furthermore, TIMER2.0 enabled us to examine the interplay between hub genes, cancer purity, and immune cells, establishing a link between hub gene expression and immune involvement in cancer progression and prognosis.

IL-6, a multifunctional cytokine, plays a crucial role in immune responses, chronic inflammatory diseases, autoimmune disorders, and oncogenesis [34,35,36,37,38]. It is notably overexpressed in tumors resistant to epidermal growth factor receptor tyrosine kinase inhibitors (EGFR-TKIs), where it suppresses the activation of NK and T cell subsets and is implicated in the TGF-β signaling pathway [39,40]. Elevated IL-6 levels are associated with increased metastasis in NSCLC [41,42]. Our findings indeed indicate a correlation between IL6 expression and the aggressive progression of lung cancer. Higher IL6 expression correlated with reduced free progression survival (FPS) in patients. Moreover, data from the TCGA database reveal a wide range of IL6 expression across various subtype tissue classifications in both normal and lung cancer tissues, suggesting that further differentiation within sample groups is necessary to elucidate the causes of tissue-specific expression variations.

Human nuclear receptors (NRs), comprising 48 members, play a crucial role in regulating cellular homeostasis and pathophysiology. The NR4A subfamily, which includes NR4A1, NR4A2, and NR4A3, is unique because these orphan receptors do not have known physiological ligands. NR4A subfamily genes are rapidly activated by various stimuli, leading to shared and distinct biological functions [43]. Notably, NR4As are often up-regulated in solid tumors, whereas their expression is typically reduced in hematological malignancies [44]. NR4A1 has been found to be overexpressed in a spectrum of solid tumors, such as breast, pancreatic, ovarian, colon, endometrial, and rhabdomyosarcomas [45,46,47,48,49]. Additionally, NR4A1 plays a critical role in the TGFβ-induced invasion of breast and lung cancer cells [50,51]. Similarly, NR4A2 has been characterized as pro-oncogenic in various solid tumor cell lines, contributing to cancer cell proliferation, survival, and metastasis [52,53]. In colon cancer, the overexpression of NR4A2 is associated with increased chemoresistance and serves as a negative prognostic indicator [54]. Recent research also suggests that NR4A2 is a negative prognostic factor in glioblastoma, with the suppression of NR4A2 leading to reduced tumor growth and invasiveness [55]. Furthermore, NR4A2 is implicated in the proliferation of lung cancer cell lines [53]. In support, this study found that the high expression of NR4A1 was associated with a shorter FPS. However, analysis of TCGA data revealed that the expression of NR4A1 and NR4A2 in LUAD was lower than in normal tissues, suggesting that NR4A1 and NR4A2 may not be suitable targets for LUAD.

The FOSB and FOS genes, part of the AP-1 transcription factor complex, regulate gene expression in response to various stimuli, including those related to cancer development. In triple-negative breast cancer, a significant reduction in FOSB promotes cell proliferation and tumor growth [56]. Similarly, in gastric cancer, abnormal FOSB expression correlates with tumor progression and poor survival; more precisely, the overexpression of FOSB suppresses cell proliferation, clone formation, and migration, while the down-regulation of FOSB promotes these processes [57]. In NSCLC, FOSB acts as a potential tumor suppressor. The up-regulation of FOSB attenuates NSCLC growth and survival both in vitro and in vivo, while the down-regulation of FOSB in NSCLC patients is correlated negatively with pathological grade [58]. In oral squamous cell carcinoma, c-Fos expression is weaker in cancerous tissues compared to normal tissues, and c-Fos mRNA expression is significantly decreased in cancer patients. Conversely, the up-regulation of c-Fos contributes to the malignant phenotype of NSCLC cells [59,60]. Although these roles are consistent with the Kaplan–Meier plot predictions in our study, the TCGA database indicates a significant down-regulation of FOSB and FOS in LUAD, with varying expression levels across tissue subtypes. Therefore, further exploration is needed to fully understand the predictive effects of FOSB and FOS on patient prognosis in NSCLC.

DUSP1 is a critical enzyme that dephosphorylates and inactivates MAPKs, thus playing a significant role in the regulation of cellular responses to stress and extracellular stimuli [61]. The modulation of DUSP1 is thought to impair tumor growth and motility as well as potentiate therapeutic sensitivity in various neoplasms through the regulation of the MAPK signaling pathway [62]. In pancreatic cancer, DUSP1 is a novel target for enhancing cell sensitivity to gemcitabine, a common chemotherapy drug. Compared with gemcitabine treatment alone, knockdown of DUSP1 with gemcitabine treatment improves animal survival, attenuates angiogenesis, and enhances apoptotic cell death [63]. In addition, the overexpression of DUSP1 is correlated with poor patient survival in ovarian cancer [64]. In NSCLC, DUSP1 promotes osimertinib drug-tolerant persistence by inhibiting MAPK/ERK signaling. Conversely, silencing DUSP1 can attenuate this resistance, inhibit cell proliferation, and enhance apoptosis in NSCLC cells [65]. Further exploration is needed to fully understand the predictive effects of DUSP1 on patient prognosis in NSCLC.

Activated through the MAPK signaling pathway, EGR1 is a transcription factor involved in cell proliferation, differentiation, invasion, and apoptosis that can either induce or inhibit the expression of its target genes [66]. EGR1 is associated with the initiation and progression of cancer via involvement in tumor cell proliferation, invasion, metastasis, and angiogenesis [66]. Abnormal expression of EGR1 has been identified in prostate and gastric cancers, thus functioning as an oncogene [67,68]. EGR1 is also known to stimulate epithelial–mesenchymal transition in NSCLC cells [66]. While EGR1 also exhibited a favorable association with overall survival (HR = 0.6, log-rank *p* = 3.3 × 10^−9^), its expression pattern in LUAD was more variable. This less distinct expression profile may limit its applicability as a therapeutic target, further research is needed to confirm whether EGR1 can be used as an early indicator of poor prognosis.

ATF3 is a stress-induced transcription factor that modulates metabolism, immunity, and oncogenesis [69]. Low levels of ATF3 expression have been associated with poor overall survival in patients with hepatocellular carcinoma (HCC). ATF3 is also expressed at low levels in multiple HCC tumor tissues and is significantly associated with clinical cancer stage and pathological tumor grade [70]. The expression levels of ATF3 have been found to correlate with cisplatin resistance in NSCLC. Higher induction of ATF3 is observed in sensitive cell lines compared to resistant cell lines following cisplatin treatment [71]. In addition, activating the ATF3 pathway induces ferroptosis in cisplatin-resistant NSCLC cells [72]. Further research is needed to fully understand the implications of ATF3 expression levels in NSCLC.

ZFP36, also known as tristetraprolin (TTP), is an RNA-binding protein that recognizes adenylate-uridylate-rich elements (AREs) in the 3′-untranslated regions of target mRNAs and promotes their degradation, thereby regulating the post-transcriptional expression of genes involved in inflammation and tumorigenesis [73]. In hepatocellular carcinoma (HCC), ZFP36 functions as a tumor suppressor by destabilizing oncogenic mRNAs such as PRC1, CCND1, and CDK6, which are implicated in cell cycle progression and tumor growth [74,75]. Reduced expression of ZFP36 in HCC is correlated with advanced tumor stage and poor prognosis. Moreover, the ZFP36/PRC1 regulatory axis has been proposed as a potential therapeutic target, as restoring ZFP36 expression attenuates tumor proliferation and enhances chemosensitivity to agents such as 5-fluorouracil [73]. Additionally, the loss of ZFP36 leads to the up-regulation of oncogenes and promotes malignant phenotypes in NSCLC [76]. Similarly, ZFP36 demonstrated a significant association with favorable prognosis in LUAD, as indicated by Kaplan–Meier analysis (HR = 0.75, log-rank *p* = 0.0053). However, its expression pattern across LUAD samples was less consistent when analyzed using UCSC Xena. ZFP36 expression levels varied considerably among different LUAD subtypes. This heterogeneity in expression reduces its specificity and limits its utility as a robust biomarker or therapeutic target compared to PECAM1. This suggests that the functional role and clinical relevance of ZFP36 in lung cancer remain largely unexplored and warrant further investigation.

PECAM1 is pivotal in angiogenesis, immune responses, and preserving endothelial barrier integrity [8]. Its aberrant expression is implicated in atherosclerosis, thrombosis, leukemia, and diverse cancers. Notably, PECAM1 overexpression has been shown to augment melanoma and breast cancer cell proliferation [10,11]. Additionally, PECAM1 phosphorylation contributes to the anoikis resistance of lung cancer cell lines [9]. Contrary to its association with cancer progression, our research reveals the diminished expression of PECAM1 in lung cancer tissues. Intriguingly, higher levels of PECAM1 correlate with extended survival among lung cancer patients. This paradox suggests a multifaceted role of PECAM1, particularly in the context of immunotherapy—a field garnering substantial interest. The infiltration of immune cells, facilitated by PECAM1, appears to thwart tumor growth [77,78,79]. Our public data mining has revealed that PECAM1 is a lowly expressed differentially expressed gene (DEG) but a pivotal hub gene that regulates a variety of immune functions and is positively associated with immunoglobulins. These findings reflect the role of PECAM1 in regulating immune surveillance and suggest that its anti-cancer properties could be associated with immune regulation. One limitation of our study is that the microarray datasets used (GSE10072, GSE33356, and GSE18842) lacked sufficient sample size by tumor stage or grade. Although external validation using TCGA data revealed consistently lower expression of the identified hub genes across various clinical stages, further studies involving stage-specific and grade-specific datasets are warranted to determine whether PECAM1 and related markers retain prognostic significance across different LUAD subtypes.

Furthermore, recent research by Varnamkhasti et al. identified ZNF334 as a gene with clinical significance in lung squamous cell carcinoma (LUSC), particularly in relation to lymph node metastasis and bypassing cellular senescence mechanisms [80]. Their study highlighted how ZNF334 dysregulation could serve as a hallmark of metastatic progression in LUSC through the activation of genes related to chromosomal instability and p53 pathway inhibition. Although our current study focuses primarily on lung adenocarcinoma (LUAD) and identifies PECAM1 as a potential prognostic biomarker, these findings collectively underscore the growing importance of subtype-specific biomarkers in NSCLC. Integrating such emerging evidence broadens the molecular landscape of lung cancer progression and reinforces the need to evaluate distinct gene signatures—including PECAM1 and ZNF334—across different histological contexts to optimize biomarker-guided strategies for early detection and treatment.

The reliance on public gene databases means the data are subject to the quality and comprehensiveness of these datasets. Additionally, while bioinformatics analyses can provide valuable insights, they cannot replace experimental validation in clinical settings. Although PECAM1 was consistently identified as a down-regulated hub gene across multiple independent datasets and demonstrated significant prognostic association, its mechanistic role and clinical applicability must be confirmed through laboratory-based studies. Future in vitro and in vivo experiments are necessary to validate its biological function in tumor progression, immune modulation, and therapeutic responsiveness to establish the therapeutic potential of the identified target genes.

## 5. Conclusions

Our study utilized a bioinformatics approach to identify and characterize genes with differential expression in lung cancer. Our findings shed light on the potential roles of these genes in lung cancer pathogenesis and their implications for prognosis and therapy. PECAM1 stands out as a gene with significant prognostic value, warranting further investigation as a potential biomarker and therapeutic target in lung cancer.

## Figures and Tables

**Figure 1 diagnostics-15-01094-f001:**
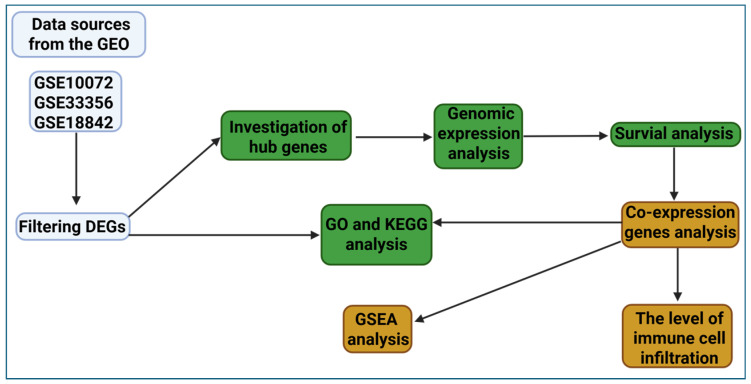
Overview of the bioinformatics workflow used in this study, including dataset selection, DEG identification, enrichment analyses, hub gene prioritization, and prognostic evaluation.

**Figure 2 diagnostics-15-01094-f002:**
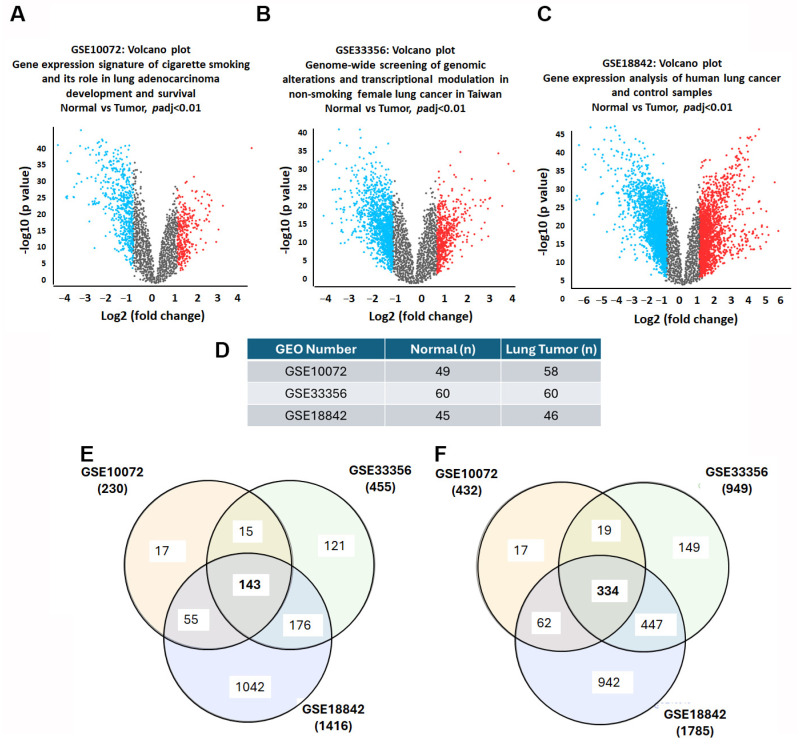
Intersecting DEGs in GEO datasets in lung cancer: (**A**–**C**) volcano plots of genes significantly up-regulated (red) and down-regulated (blue) in lung tumors compared to normal tissue; (**D**) sample sizes: GSE10072 (49 normal and 58 lung cancer tissues), GSE33356 (60 normal and 60 lung cancer tissues), and GSE18842 (45 normal and 46 lung cancer tissues); (**E**,**F**) Venn diagrams of overlapping up-regulated and down-regulated DEGs.

**Figure 3 diagnostics-15-01094-f003:**
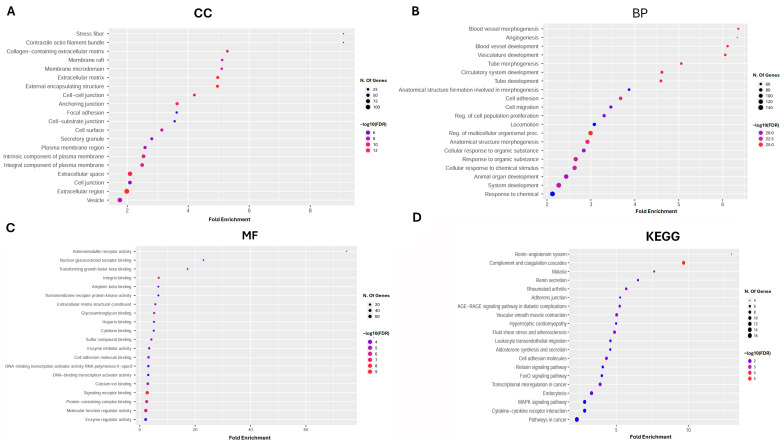
Functional roles of DEGs in gene ontology analysis (color marks false discovery rate (FDR): (**A**) cellular component (CC); (**B**) biological process (BP); (**C**) molecular function (MF); and (**D**) Kyoto encyclopedia of genes and genomes (KEGG).

**Figure 4 diagnostics-15-01094-f004:**
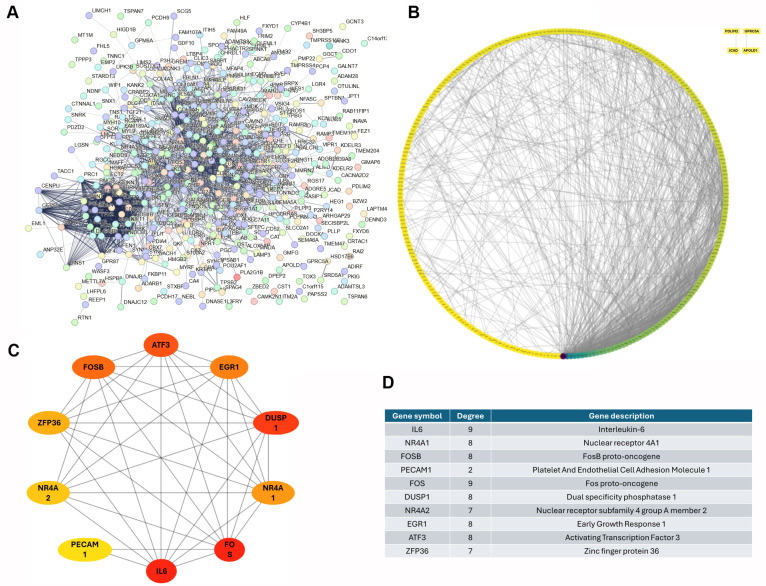
Top 10 down-regulated hub genes in lung cancer: (**A**) The protein–protein interaction (PPI, *p*-value was <1.0 × 10^−16^) generated by STRING; (**B**) DEG clusters generated by Cytoscape; (**C**) top 10 hub genes screened using Cytoscape; (**D**) hub genes ranked by degree. Note: IL6, interleukin 6; NR4A1, nuclear receptor 4A1; FOSB, FosB proto-oncogene; PECAM1, platelet and endothelial cell adhesion molecule 1; FOS, Fos proto-oncogene; DUSP1, dual-specificity phosphatase 1; NR4A2, nuclear receptor subfamily 4 group A member 2; EGR1, early growth response 1; ATF3, activating transcription factor 3; ZFP36, zinc finger protein 36.

**Figure 5 diagnostics-15-01094-f005:**
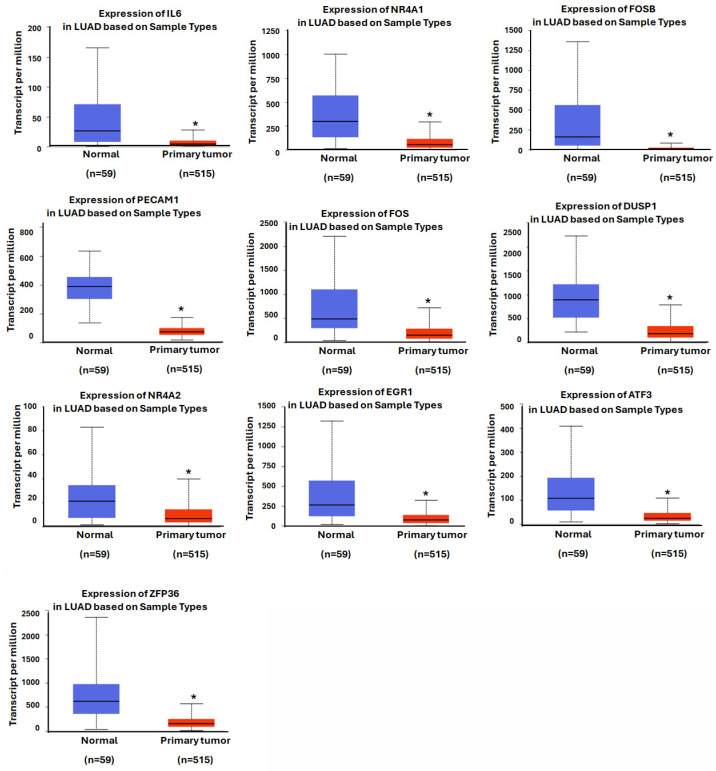
Comparison of top 10 hub gene expressions in LUAD and normal tissues in TCGA database. *: *p*< 0.05.

**Figure 6 diagnostics-15-01094-f006:**
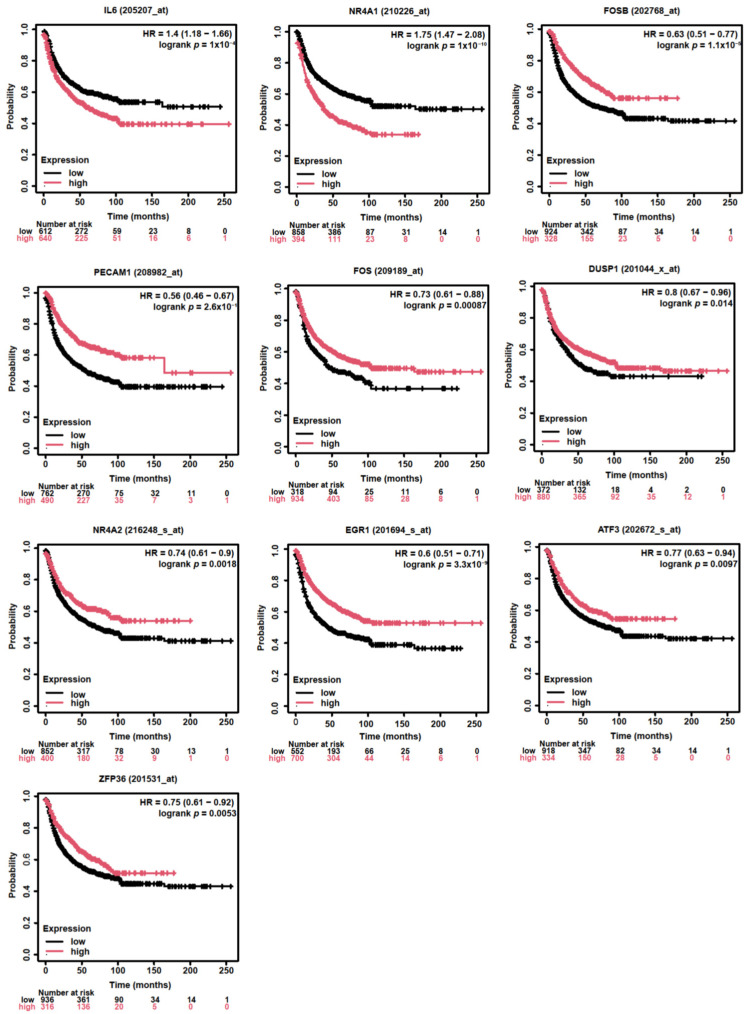
Assessment of the first-progression survival of hub genes: The First-progression survival (FPS) of top 10 hub genes obtained from Kaplan–Meier analysis (HR = hazard ratio; and red and black, respectively, high and low hub gene expression).

**Figure 7 diagnostics-15-01094-f007:**
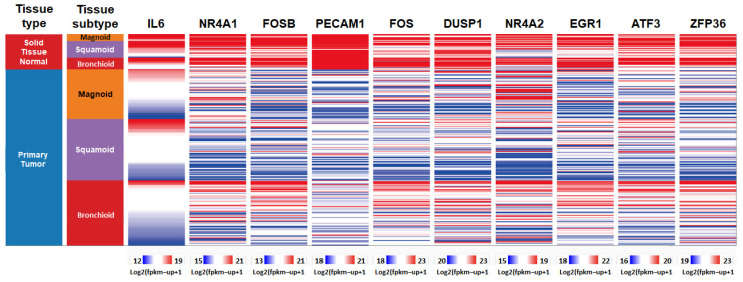
Expression of hub genes in lung cancer subtypes: UCSC Xena analysis of expression of hub genes in normal and pathological lung tissues with sub-tissues magnoid (orange), suqamoid (purple), and bronchioid (red).

**Figure 8 diagnostics-15-01094-f008:**
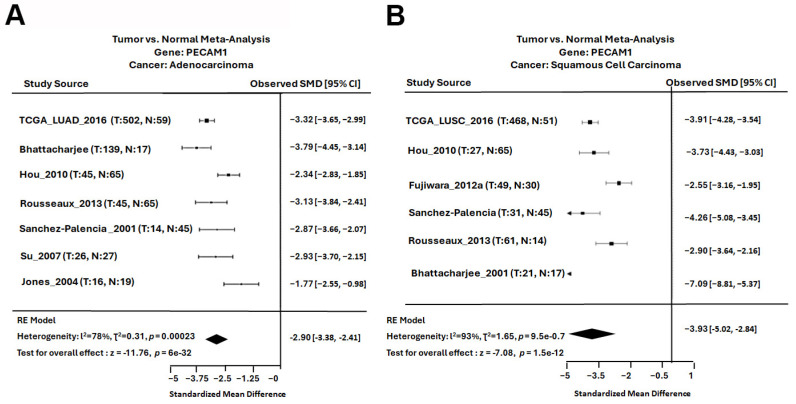
Meta-analysis of PECAM1 in NSCLC: (**A**) meta-analysis of PECAM1 expression in LUAD and (**B**) meta-analysis of PECAM1 expression in LUSC. The random effects model was applied when determining the standardized mean difference (SMD), which is represented by the black squares for each dataset. The varying sizes of the squares reflect the weight of each study, the horizontal lines represent the 95% confidence interval (CI) of each study, and the diamond represents the overall effect size.

**Figure 9 diagnostics-15-01094-f009:**
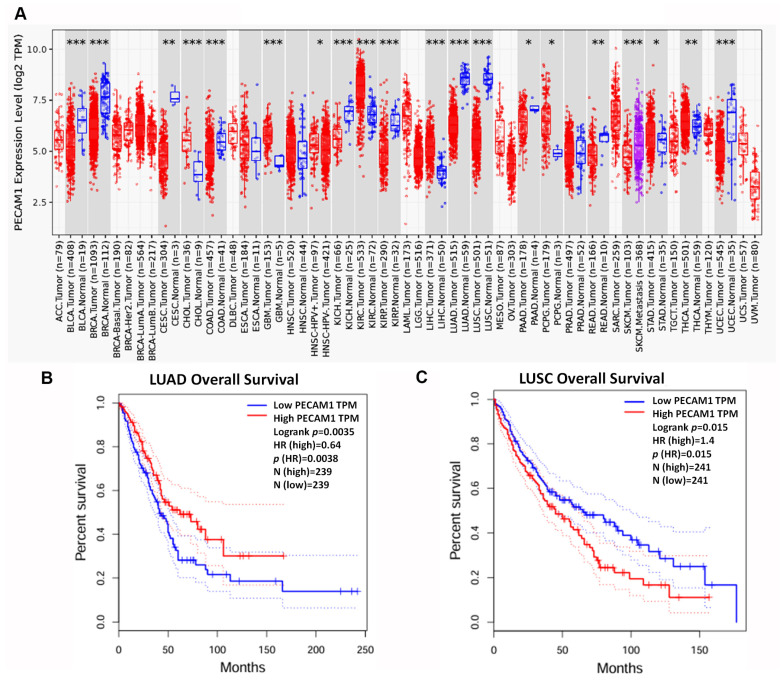
Expression of PECAM1 in pan-cancer: (**A**) expression of PECAM1 in LUAD and various cancer types in the TCGA database analyzed using TIMER2.0; (**B**,**C**) Kaplan–Meier plots of PECAM1 expression level and overall survival in LUAD and LUSC patients. *: *p* < 0.05; **: *p* < 0.01; ***: *p* < 0.001. Dotted line represented 95% CI.

**Figure 10 diagnostics-15-01094-f010:**
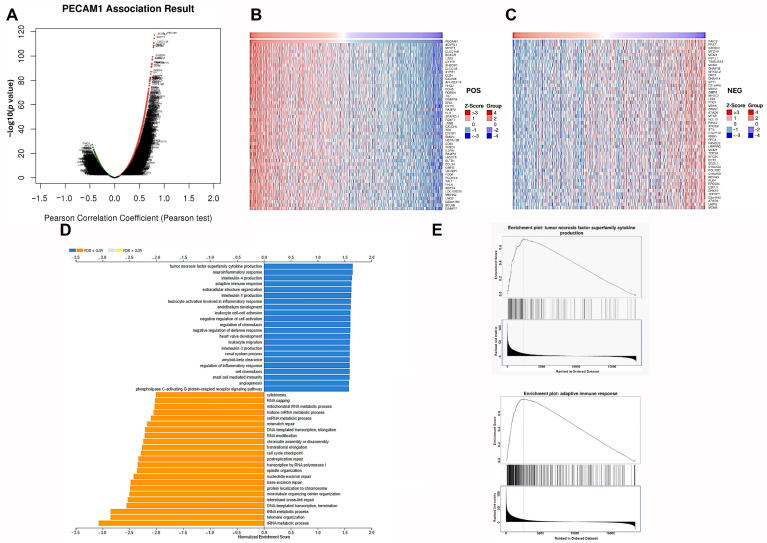
Role of PECAM1 co-expressing genes: (**A**) volcano map of co-expressed genes linked to PECAM1 expression in LUAD from TCGA datasets; (**B**,**C**) heat maps of top 50 co-expressed genes positively and negatively correlated with PECAM1 in LUAD from TCGA datasets; (**D**) gene ontology analysis of PECAM1 co-expression genes, focusing on biological processes and molecular functions; (**E**) GSEA of PECAM1 co-expression genes, focusing on TNF superfamily cytokine production and adaptive immune response.

**Figure 11 diagnostics-15-01094-f011:**
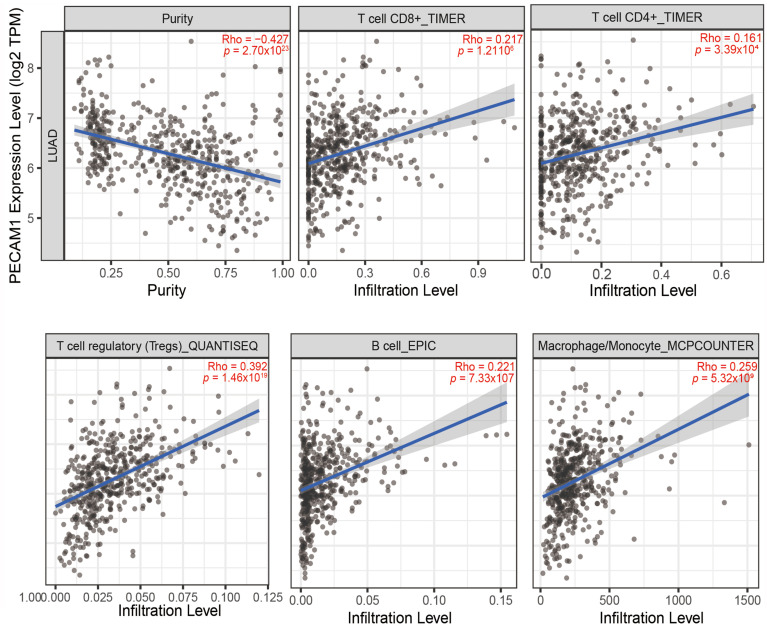
Correlations among PECAM1, tumor purity, and immune cells (TPM, transcripts per million).

## Data Availability

The data presented in this study are available on request from the corresponding author.

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
