# Peer review of "Identification of PECAM1 as a Prognostic Biomarker for Lung Adenocarcinoma"

_diagnostics, 2025, doi:10.3390/diagnostics15091094_

Round 1
Reviewer 1 Report
Comments and Suggestions for Authors
The authors used Transcriptomic data and analyzed differentially expressed genes in clinical cases of lung adenocarcinoma. I have some comments which needs to be addressed to improve this study.
- Did the authors consider the grades or stages of those patients with lung adenocarcinoma? In all of these cases, will their biomarker be consistent in terms of quantities/ or levels of expression? They need to justify and elaborate this further.
2. The authors have a comprehensive discussion however, a flowchart or an infographic may be necessary to improve .
3.The authors need to mention limitations of their study and what possible in vitro/in vivo experiments in the future which can improve this type of study.
Reviewer 2 Report
Comments and Suggestions for Authors
In this study, the authors bioinformatically investigated the hub genes involved in lung cancer. The following are suggested to improve their work:
-In the summary results section, statistical parameters such as P-values ​​should be included.
-In the introduction section, a better and more comprehensive background of previous bioinformatics and laboratory studies on lung cancer should be provided. The following articles are suggested if the authors deem it appropriate:
1.
Genes whose expressions in the primary lung squamous cell carcinoma are able to accurately predict the progression of metastasis through lymphatic system, inferred from a bioinformatics analyses
2.
LINC00894, YEATS2-AS1, and SUGP2 genes as novel biomarkers for N0 status of lung adenocarcinoma
-Complete information is not provided about the characteristics of the GSE examined in this study.
-Considering that different GSEs have been checked, is there any role for batch effect? ​​How did you remove it?
-Explain more about the basis for obtaining the ten hub genes.
-In the discussion section, you can use newly published articles such as
Implications of ZNF334 gene in lymph node metastasis of lung SCC: potential bypassing of cellular senescence
-Given that your results are based on bioinformatics analyses, how confident are you that this gene will receive the same confirmation in the laboratory?
Reviewer 3 Report
Comments and Suggestions for Authors
Dear Editor and Authors,
It was with interest that I read and evaluated the manuscript titled "Identification of PECAM1 as a Prognostic Biomarker for Lung Adenocarcinoma" by Dr. Shih-Sen Lin and colleagues from Taiwan.
In this work the authors utilized 3 microarray datasets (GSE10072, GSE33356, and GSE18842) to identify differentially expressed genes (DEGs) associated with lung cancer prognosis and immune infiltration. Gene Ontology (GO) as well as Kyoto Encyclopedia of Genes and Genomes (KEGG) pathway analyses were performed to elucidate the biological roles of the DEGs in lung cancer. Kaplan-Meier Plotter and TIMER2.0 were used to explore the prognostic significance of these hub genes. From the data 477 DEGs were identified from all three datasets. Ten downregulated hub genes were identified of which
PECAM1, a cell adhesion molecule which is expressed on the surface of endothelial cells and which plays a significant role in angiogenesis, immune responses, and maintaining endothelial barrier integrity was significantly downregulated in lung cancer samples and thus the authors believe has prognostic value. Because of PECAM1's association with patient survival and immune infiltration the authors suggest that it could serve as a valuable target for early cancer detection and for the development of immunotherapeutic drugs.
The work is well set up and follows appropriate and logical methodology. The authors have performed a good analysis of the data sets utilizing established techniques and were able to produce resonable outcomes. The prognostic association utilizing linked patient survival data was also quite well performed.
Overall the manuscript is well written and set up, with good flow and clear language wheras the graphs included are informative and comprehensive.
In terms of clinical significance, I am unsure I agree with the authors if PECAM1's association will prove as improtant as the authors imply in their discussion. This is a matter to be seen but I feel that as a finding it does have merit to be presented in the clinical/oncological community at least as a potential target for investigation.
In conclusion, I am happy to allow this work to go forward. Kindest regards to all.
Round 2
Reviewer 1 Report
Comments and Suggestions for Authors
The manuscript has been improve0d.